# Heat stress disrupts early development and photosymbiosis in *Cassiopea* jellyfish

**Celeste Robinson**[1], **Jingchun Li**[1,2], **Ruiqi Li**[1,2¤a]*, **Viridiana Avila-Magaña**[1¤b]*

**1** Department of Ecology & Evolutionary Biology, University of Colorado Boulder, Boulder, Colorado, United States of America, **2** Museum of Natural History, University of Colorado Boulder, Boulder, Colorado, United States of America

¤a Current Address: Department of Biological Sciences, University of Southern California, Los Angeles, California, USA
¤b Current Address: Division of Biosphere Sciences and Engineering, Carnegie Science, Pasadena, California, USA
* Ruiqi.Li@Colorado.edu (RL); viridianaavilamagana@gmail.com (VA)

## Abstract

Photosymbioses between Cnidarians and algae are widespread in marine ecosystems. The jellyfish *Cassiopea-Symbiodinium* symbiosis serves as a valuable model for studying host-symbiont interactions in photosymbiotic organisms. Despite its ecological similarity to coral symbiosis, the effects of rising sea surface temperatures on *Cassiopea* symbiosis, particularly during early developmental stages, remain unexplored. By exposing *Symbiodinium* cultures to heat stress and subsequently using these symbionts to colonize jellyfish polyps under ambient and elevated temperature conditions, we study the impact of heat on microbe-stimulating strobilation. We observed a significant reduction in chlorophyll concentration in heat-stressed *Symbiodinium* algae. Polyps colonized with these symbionts exhibited delayed strobilation under ambient conditions and failed to undergo strobilation under continued heat stress. Additionally, we found abnormal ephyra morphology and increased rates of asexual reproduction under heat stress. Our findings suggest that ocean warming may disrupt critical stages of *Cassiopea* strobilation and development, ultimately threatening their population stability under warming marine environments.

## Introduction

Photosymbiotic relationships between marine invertebrates and photosynthetic algae play a crucial role in tropical marine ecosystems, driving primary production, nutrient cycling, and biodiversity [1]. Among these, cnidarian photosymbiosis is particularly significant, as it forms the foundation of one of the most productive habitats on Earth—coral reefs. However, rising sea surface temperatures due to climate change threaten these associations by disrupting symbiotic interactions [2,3]. Understanding the effects of thermal stress on these partnerships is essential for predicting broader

**Data availability statement:** All relevant data are within the manuscript and its Supporting Information files.

**Funding:** This work is funded by a Packard Fellowship for Science and Engineering (2019–69653) to JL. The funders had no role in study design, data collection and analysis, decision to publish, or preparation of the manuscript.

**Competing interests:** No authors have competing interests.

ecological consequences. In recent years, research has largely focused on corals due to their vulnerability to climate change, but the impacts of heat stress on other photosymbiotic cnidarians remain understudied.

Among other photosymbiotic cnidarians, the upside-down jellyfish, *Cassiopea xamachana*, is a particularly interesting model due to its symbiosis with Symbiodiniaceae dinoflagellates, which not only provides energy but also regulates its strobilation. *C. xamachana* has a complex life cycle involving both sexual and asexual reproduction [4,5]. Polyps reproduce asexually through stolon budding, generating planuloid buds [6], while sexually mature medusae produce planula larvae that settle and develop into new polyps [7]. Unlike most jellyfish, where sexual reproduction is influenced by seasonal temperature changes [8], *C. xamachana* requires colonization by Symbiodiniaceae symbionts to initiate strobilation. Without these symbionts, polyps remain indefinitely at the benthic stage and fail to undergo strobilation [7]. Once infected, polyps begin strobilation, producing ephyrae that develop into adult medusae (Fig 1).

Relying on symbionts for strobilation may make *C. xamachana* particularly vulnerable to rising temperatures. Although its primary symbiont, *Symbiodinium microadriaticum*, is relatively thermotolerant compared to other Symbiodiniaceae species [9], prolonged heat stress can still lead to pigment loss, photosynthetic dysfunction, and even photoinhibition in *S. microadriaticum* [10,11]. In Scleractinians, *Acropora digitifera* larvae exhibited higher mortality or prematurely underwent metamorphosis under heat stress [12]. The impact of heat stress on *C. xamachana* development remains unexamined, but studies have shown that ocean acidification can induce developmental abnormalities, such as causing an inverted bell shape [13]. Investigating how heat stress affects *C. xamachana*'s reproduction and development is particularly valuable, as it provides insights into both the consequences of nutritional symbiosis breakdown and the cascading effects of heat stress on strobilation cues.

In this study, a heat-stress experiment was conducted on *C. xamachana* polyps and their symbiont, S. *microadriaticum*, to examine how elevated temperatures influence their symbiotic relationship, particularly in terms of development and reproduction. A combination of morphological observations, symbiont density quantification, and chlorophyll concentration analysis was used to quantitatively and qualitatively assess the effects of heat stress on *C. xamachana* strobilation timing, ephyrae development and asexual reproduction patterns. By addressing these questions, this study provides insights into how ocean warming may disrupt cnidarian-algal symbiosis and impact *C. xamachana* populations in a warmer ocean.

## Materials and methods

### Materials

*C. xamachana* polyps were originally collected from Key Largo, Florida Keys. A clonal polyp line was established in Mónica Medina's lab at Pennsylvania State University. A subculture was later established at the University of Colorado Boulder and used in this study. These clonal polyps were maintained in filtered artificial seawater tanks at a constant temperature of 26°C, and a salinity of 35 ppt following the protocol in [14].

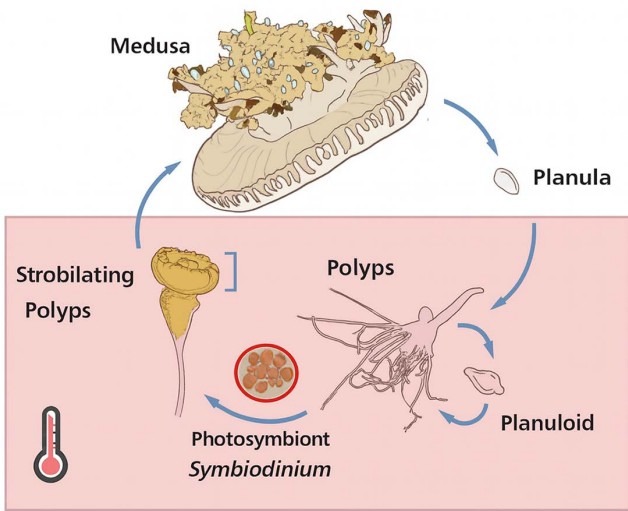

**Fig 1. Life cycle of *Cassiopea xamachana*.** *C. xamachana* can reproduce sexually and asexually, with strobilation induced by Symbiodinium Colonization. Polyps can regenerate after strobilation. The red box highlights the life stage examined in this study. Figure was modified from Ohdera 2018.

Polyps were fed every other day with the San Francisco strain of *Artemia* brine shrimp (Brine Shrimp Direct). The KB8 strain of *Symbiodinium microadriaticum* [15] was cultured in 0.1 M F/2 media. Cultures were maintained in a temperature and light-controlled incubator (Percival Scientific, Perry, USA) at 26°C, 140 µmol quanta m$^{-2}$s$^{-1}$, under a 12:12 h light-dark cycle, at a density of ~4.26 million cells/mL.

## Ethics statement

This study involved the upside-down jellyfish *Cassiopea xamachana*, which are non-cephalopod invertebrates. In accordance with institutional and national regulations, research on these animals does not require approval from an Institutional Animal Care and Use Committee (IACUC) or equivalent animal ethics board.

## Experiment design

To assess the impact of thermal stress on *C. xamachana* development and reproduction, we established three experimental groups using *S. microadriaticum* to infect *C. xamachana* polyps (Fig 2). A temperature of 34°C was chosen for heat treatment because that is the upper limit of relevant ecological broad temperature range (26–34°C) for *C. xamachana* ([16–18]), has been reported to support the maintenance of homeostatic bell pulsation within that range [18], and marks the onset of irregular pulsation and early photosymbiotic breakdown in medusae [19]. Symbionts were pre-exposed to 34°C for 8 days before inoculation, and chlorophyll content was measured on days 6 and 8 using a NanoDrop spectrophotometer (Thermo Scientific, Waltham, USA) following Klein et al. (2019) [20]. In addition, algal cell density was quantified using a Countess II FL automated cell counter, by staining the cells with Trypan blue to determine cell viability (Invitrogen, Waltham, USA). Inoculations were conducted with symbionts in their exponential growth phase at $1 \times 10^6$ cells ml$^{-1}$ following Medina et al. 2021 [21]. The three experimental groups were termed control group, heat group, and recovery group respectively. The control group consisted of polyps and symbionts both raised and maintained at 26°C. The heat group was infected with symbionts cultured at 34°C and maintained at 34°C. In the recovery group, polyps were infected with symbionts cultured at 34°C but maintained at 26°C after inoculation. Inoculations were conducted in three 6-well plates, one per experimental group, with 12–13 polyps in 4 of the 6 well plates, leaving two plates empty (Fig 2). 12 hours post

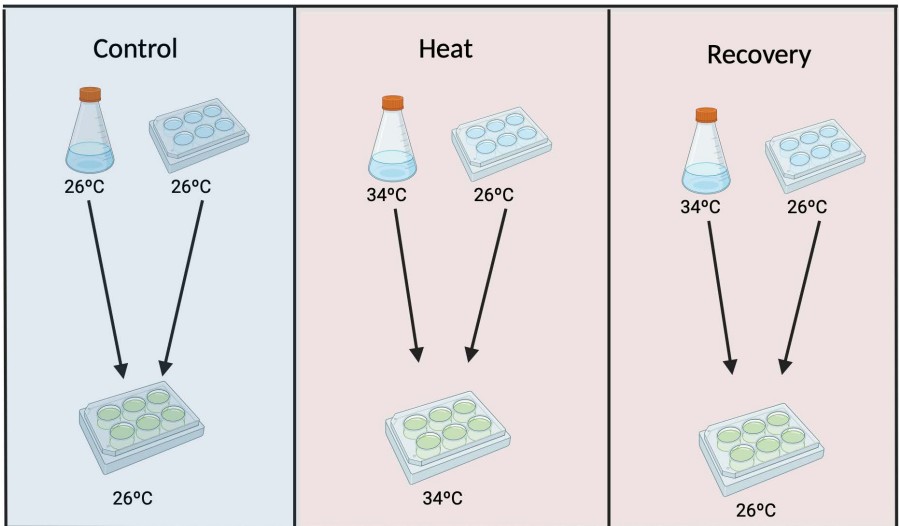

**Fig 2. Experimental design to assess the impact of thermal stress on *C. xamachana* development.** Three groups were used: 1) Control – polyps and symbionts both cultured and maintained at 26°C. 2) Heat – polyps infected with symbionts cultured at 34°C and maintained at 34°C; 3) Recovery – polyps infected with symbionts cultured at 34°C but maintained at 26°C; and Two separate temperature and light-controlled incubators (Percival Scientific, Perry, USA) were used. With equal light intensity and duration, one incubator was set to 26°C and the other to 34°C, used for the corresponding groups throughout the experiment.

inoculation, we removed all water from the 6 well plates using a pipette and rinsed them to remove any spent media after inoculation, replenishing the wells with filtered sea water and resumed normal feeding and cleaning procedures.

## Measurements

To evaluate the effect of heat stress on host strobilation, the number of polyps undergoing strobilation was observed every other day during normal rearing, and when strobila were present.

Photos of ephyrae were captured using a Micromaster M002 microscope (Fisher Scientific, Waltham, USA) to identify potential morphological abnormalities. To quantify algal cell density within the ephyrae, five ephyrae from the control group (5 mm diameter, 19.63 mm$^3$) and ten from the recovery group, including five inverted bell-shaped (3 mm diameter, 7.07 mm$^3$) and five normal ephyrae (4 mm diameter, 12.57 mm$^3$), were collected fifteen days after the onset of strobilation. Samples were flash-frozen, then thawed, macerated in 30 µL of seawater with a pestle until homogenized, then 20 µL of seawater was added to rinse the pestle, and the solution was vortexed for 20 minutes. The solution was centrifuged at 4,000 rpm for 10 min, and the supernatant was removed, the pellet was resuspended in seawater. Cells were stained with Lugol, and 10 µL of the homogenized suspension was loaded onto a hemocytometer, algal cells were counted under a Fisher Scientific Micromaster M002 microscope at 10 × magnification, cell counts were normalized to cells per ephyra volume. At least four replicates were counted per sample.

To assess the impact of heat stress on host asexual reproduction, the number of planuloid buds in each well was recorded throughout the experiment.

## Statistical analyses

The effect of heat stress and morphological forms on *in hospite* algal cell density was analyzed by comparing algal cell density in ephyrae raised under normal temperatures to those exposed to heat stress (normal and inverted forms). T-tests were performed to evaluate statistical significance between groups using R [22]. Plots were generated using ggplot2 package in R.

## Results and discussion

### Heat-stressed *Symbiodinium* prevented *Cassiopea* strobilation

Our experiments revealed a significant delay in symbiont-induced strobilation when symbionts were pre-exposed to thermal stress in the recovery group. While control polyps completed strobilation by day 19, full strobilation in the recovery group was delayed until day 36 (Fig 3). In contrast, no strobilation occurred under continuous heat stress. Although it is possible that *Cassiopea* is inherently unable to strobilate under prolonged heat stress, the significant delay observed in the recovery group suggests that physiological changes in the symbionts are more likely to be responsible for the disruption in strobilation. Symbiont cell viability did not show a significant difference between the heat and control groups (Fig 4a), with average cell density ranging from $3.27 \times 10^6$ cells/mL to $4.15 \times 10^6$ cells/mL (SD = 1.196 and 0.427, respectively). However, chlorophyll a quantification indicated a substantial reduction in pigment concentration under thermal stress (Fig 4b). Chlorophyll a concentration for the control group were 0.016 g/L and 0.031 g/L for day 6 and 8 respectively, while those for the heat group were 0.004 g/L and 0.011 g/L. Heat-stressed Symbiodinium cultures were also visibly lighter in color compared to the control cultures, supporting lower pigment content. (Fig 4c). These findings suggest that heat stress alters symbiont physiology in a way that may impair their ability to induce timely strobilation in *Cassiopea*. However, only one measurement per day was taken, so further replication would be necessary to further validate these results.

Strobilation cues in jellyfish vary widely among species and can be influenced by environmental factors such as temperature, salinity, and light exposure. These patterns are typically inferred by correlating environmental variables with observed strobilation events and are often validated through laboratory studies. For example, *Aurelia aurita* strobilates after prolonged cold periods [23,24], whereas species like *Cephea cephea* and *Rhopilema verrilli* respond to elevated temperatures [24,25]. In *Aurelia labiata*, strobilation has been associated with high-temperature and high-light conditions in natural settings [26]. In *Cassiopea xamachana*, however, this transition is dependent on successful colonization by endosymbiotic *Symbiodinium* spp., which acts as an essential developmental driver [7,27]. The presence of these dinoflagellates appears to be a critical environmental and physiological cue required to trigger the transition from the polyp stage (scyphistoma) to the strobila stage, ultimately leading to the release of juvenile medusae (ephyrae). In brief, the transition is induced when symbionts are ingested and phagocytosed by gastrodermal cells, migrating basally within three days [7,28]. By day eight, symbiont-containing host cells relocate to the mesoglea and proliferate [7]. Strobilation occurs around three weeks post-colonization, influenced by symbiont species, temperature, and nutrition [29]. There is some evidence suggesting that metabolic products derived from the symbiont, specifically carotenoids, may be important for initiating *C.*

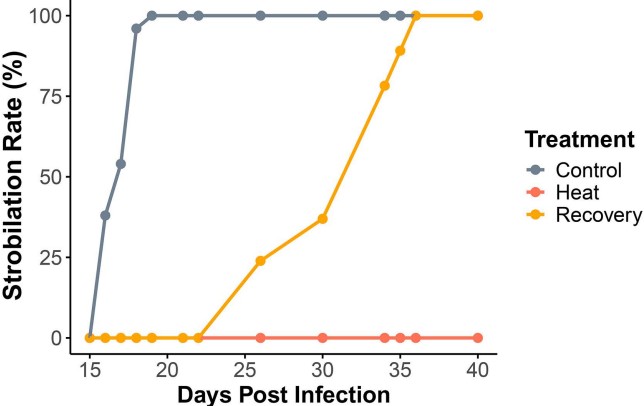

**Fig 3. Percentage of polyps undergoing strobilation over time in each treatment group.** Control group completed strobilation by day 19, while strobilation in the recovery group was delayed until day 36. No strobilation was observed in the heat group. N = 50.

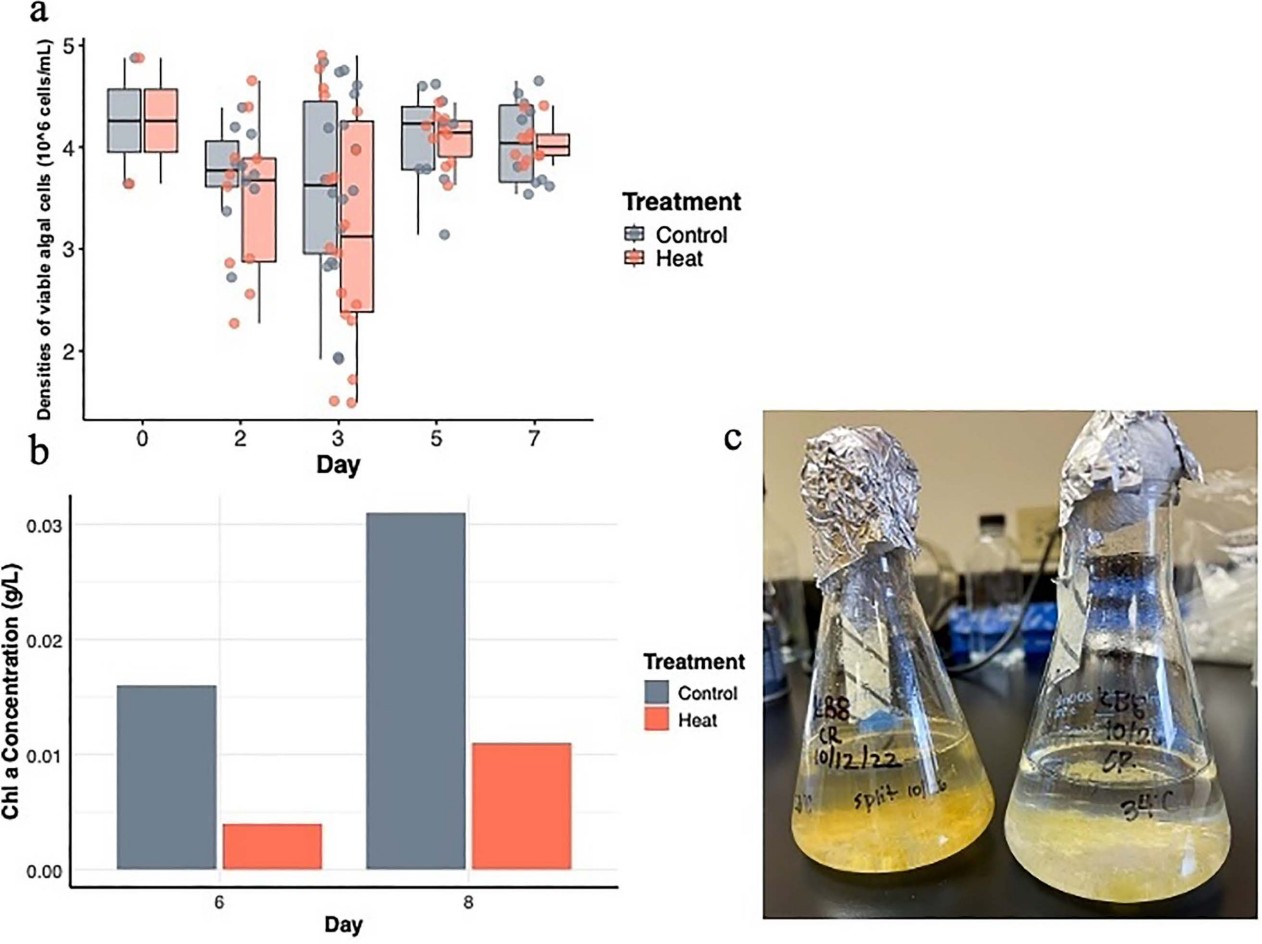

**Fig 4. Effects of heat stress on *S. microadriaticum*. (a)** Symbiont cell density did not differ significantly across treatments. **(b)** Chlorophyll content was significantly reduced under thermal stress. **(c)** Symbionts exposed to heat (right) appeared visibly lighter in color compared to the darker brown control cultures (left).

*xamachana* development [16], although it is unclear whether additional metabolites or a combination of other factors regulate and trigger host development.

The impact of thermal stress on *Symbiodinium* is complex, affecting various physiological and molecular processes. Transcriptomic studies have revealed widespread but subtle gene expression changes in *Symbiodinium* under thermal stress, affecting antioxidant networks, chaperone proteins, photosynthesis activities, and metabolic pathways [30]. Heat stress can also lead to photoinhibition by impairing photosystem II (PSII) [11]. However, impaired photosynthesis alone is unlikely to be the direct cause of delayed or halted strobilation in *Cassiopea*. Since carotenoid metabolites are believed to play a role in inducing *Cassiopea* strobilation [16] and previous studies have shown that thermal stress leads to a loss of pigments in the Light Harvesting Complex, resulting in reduced chlorophyll a and carotenoid levels in *Symbiodiniaceae* [31], the observed loss of symbiont pigmentation may explain the effect of heat stress on strobilation. Other studies have shown that indoles can induce strobilation in aposymbiotic *Cassiopea andromeda* polyps, suggesting that certain chemical compounds may serve as direct triggers for *Cassiopea* strobilation [23,25,32]. Metabolomic studies on *Symbiodiniaceae* under heat stress have identified distinct metabolic changes [33]. Further research on the *Cassiopea* holobiont is necessary to pinpoint additional photosymbiont metabolites that contribute to strobilation.

## Altered algal density and developmental abnormalities in ephyrae with symbionts exposed to heat

Our results showed that 90% (45 out of 50) of ephyrae from the recovery group exhibited an inverted bell shape and were significantly smaller (~3 mm) compared to control ephyrae (~5–6 mm) (Fig 5a). In addition, ephyrae with an inverted bell shaped contained significantly more symbiont cells ($7.0 \times 10^4$ cells/mm³, SD = 2.20) than normal-shaped ephyrae ($2.32 \times 10^4$ cells/mm³, SD = 1.39) within the recovery group ($p < 0.001$, Fig 5b). While the ephyrae in the control group had a higher algal density ($5.22 \times 10^4$ cells/mm³, SD = 1.14) than the recovery group with a normal morphology ($p < 0.001$), the

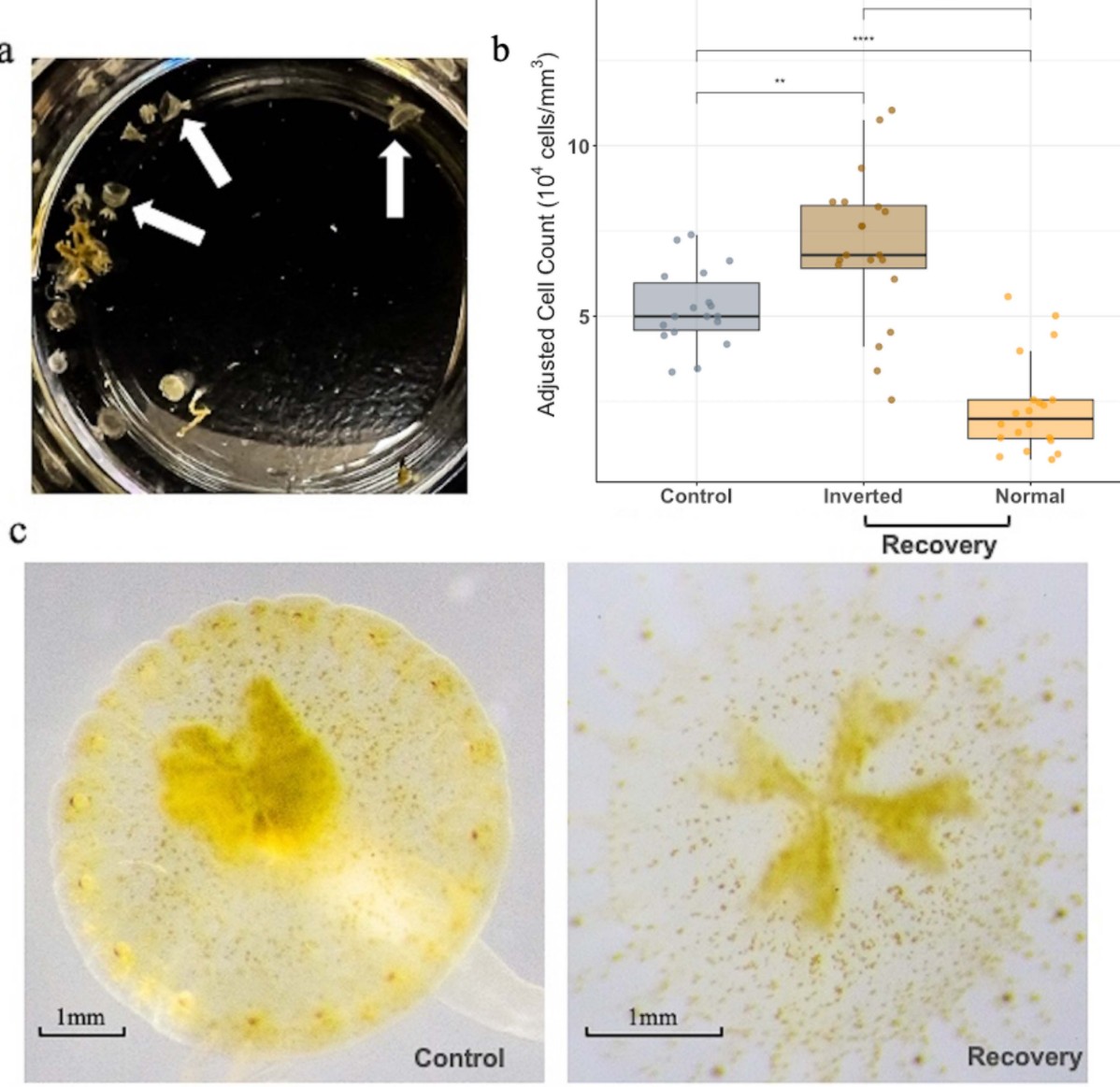

**Fig 5. Effects of heat stress *on C. xamachana* developmental morphology and symbiont density. a)** Most ephyrae in the recovery group displayed an inverted bell shape as shown in the photo. **b)** Inverted bell shape ephyrae had significantly higher symbiont densities than normal-shaped ephyrae within the recovery group **and** than in the control group. **c)** Visual comparison shows reduced symbiont density in ephyrae from the recovery group with a normal morphology compared to the control.

ephyrae from the control group had a lower algal cell density than the ephyrae with an inverted bell shape morphology from the recovery group ($p < 0.05$).

While photosynthesis may not be the direct cue for host strobilation, it is essential for proper *Cassiopea* development. Aposymbiotic *Cassiopea* polyps induced to strobilate by chemical cues often exhibit developmental abnormalities, including reduced ephyra size, malformed sensory organs, crimped marginal lappets, delayed polyp recovery after strobilation, or, in some cases, death [25]. Our results corroborated these findings, indicating that a healthy onset of photosymbiosis is a prerequisite for normal ephyra development. The most plausible mechanistic explanation is that only a stable symbiotic relationship can provide the energetic foundation required for proper morphogenesis. The importance of stable photosymbiosis for cnidarian host growth, as well as the detrimental effects of heat stress in both adult corals and *Cassiopea*, are well documented: under thermal stress, algal symbionts often lose photosynthetic efficiency [3] and retain a greater proportion of photosynthate for their own metabolism [34]. Consequently, the host receives insufficient translocated carbon to support development. This metabolic shortfall forces the host to shift from anabolic assimilation of carbon to catabolic consumption of endogenous reserves such as amino acids, a process linked to reductions in adult body size and mortality in *Cassiopea*, as well as bleaching in corals [19,34]. Such carbon limitation is likely to be even more acute in small-bodied ephyrae with minimal energy reserves, ultimately constraining growth and impairing proper morphogenesis, demonstrated by the ephyrae from the recovery group, with both normal and inverted morphology.

Why then, do ephyrae with bell abnormalities have a higher symbiont density than normal-shaped ones in the recovery group? Heat stress can alter the host-symbiont stability, potentially leading to a breakdown in mutualism. A likely driver of this paradox is the nutrient balance shift discussed above: when hosts become carbon-limited, they catabolize their own amino acids, greatly increasing ammonium (nitrogen source) availability to the symbionts, thereby fueling unchecked algal proliferation [34]. Under such conditions, symbionts proliferate more rapidly within the host and exhibit higher dispersal rates while simultaneously reducing host growth and reproduction, and lead to increase susceptibility of bleaching [34]. Thermal stress has also been documented as a disruptor of host development and immune function [35], which may further exacerbate uncontrolled symbiont growth and may strains the sensitive balance in the holobiont. This is consistent with our findings, as ephyrae exhibiting abnormal morphologies (inverted bell shape) had higher concentrations of algal cells (Fig. 5C). We noticed that the inverted bell-shaped ephyrae from the recovery group appeared darker and smaller in size when compared to the normal shaped recovery group ephyra (Fig. 5C). We speculate that the hyperpigmentation, ephyra size reduction, and shape abnormalities may represent a terminal phase, in which the photosymbionts begin to uptake host tissue in an uncontrolled fashion, leading to a symbiosis disbalance. The extent to which the host can recover its original morphology and photosymbiont density content remains unknown. However, we observed higher mortality rates in those hyperpigmented inverted-bell shaped ephyras (data not shown). Notably, similar abnormalities of the bell, such as inversion, have been observed in *Cassiopea* raised under severe seawater acidification (pH = 7.0) [11]. Further research is needed to elucidate the mechanisms underlying these developmental abnormalities and their broader ecological implications.

## Increased asexual reproduction in polyps infected with heat-stressed algae

Continuous heat stress prevents *Cassiopea* polyps from strobilating but does not alter their ability to reproduce asexually. During the polyp stage, *Cassiopea* can still reproduce through budding [36]. In our experiment, polyps infected with heat-stressed algae exhibited an increase in planuloid bud production compared to the control group. The mean number of buds per well on day 5 and day 9, respectively was as follows: 10 ± 1.8, 7.25 ± 3.8 in the control group, 14.5 ± 5.8, 18 ± 4.7 in the heat-stressed group, and 17 ± 2.6, 22.25 ± 1 in the recovery group. Both heat and recovery groups showed a higher overall number of buds (Fig 6), suggesting that heat stress may stimulate asexual reproduction.

Jellyfish reproduction often follows a boom-and-bust cycle [37], where rapid increases in reproductive output are followed by population crashes. The underlying mechanisms remain unclear, but it is possible that *C. xamachana* polyps

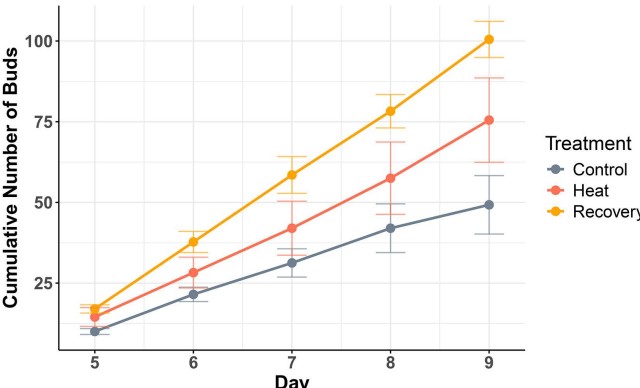

**Fig 6. Asexual reproduction via budding over time in each treatment group.** Polyps infected with heat-stressed symbionts (heat and recovery groups) produced more planuloid buds compared to the control.

under heat stress allocate their limited energy and nutrients toward offspring production as a survival strategy. Similar stress-induced reproductive responses have been observed across the animal kingdom when resources are scarce or environmental pressures intensify [38], suggesting that increased planuloid budding may be a stress response triggered by physiological changes from thermal stress.

Many jellyfish are capable of rapidly proliferating and blooming in response to rising temperatures [17,39]. In addition to increased asexual reproduction, they may also shift from asexual reproduction to strobilation, presumably to escape unfavorable conditions and facilitate habitat expansion through sexual reproduction [39]. However, in *C. xamachana*, strobilation is tightly linked to the presence of a specific photosymbiont, potentially limiting its capacity to respond quickly to environmental change unless a compatible algal partner is present. Why, then, would photosymbiotic jellyfish evolve reliance on symbionts for such a key life history transition? This dependency raises the possibility that symbiont establishment functions as a developmental checkpoint, ensuring that environmental conditions can support a stable symbiotic association, and thus medusa survival and dispersal. Supporting this idea, microbes are known to act as early indicators of environmental shifts in animal hosts [40], and animals have evolved systems like the innate immune response to detect and respond to microbial cues. These immune pathways have been co-opted for developmental regulation in some species [41] and may underlie the link between symbiosis and strobilation in *C. xamachana*. Whether this reliance on a photosymbiont ultimately provides an adaptive advantage by aligning symbiosis success with environmental suitability remains to be tested.

## Potential environmental factors impacting *Cassiopea* life-history

Thayer et al (2022) [13] reported that acidification may cause an inverted bell shape in *C. xamachana.* It is possible that other environmental factors and specifically heat stress can cause similar changes in morphology as we observed in this study, which may indirectly lead to a symbiosis impairment. We do not discard the possibility that heat stress could impact developmental processes independently of the symbiotic state. Further research needs to be done to understand the interaction of biotic and abiotic stressors on host morphological and physiological changes.

In our study, we have chosen a high temperature, i.e., 34° C to simulate an extreme heat stress scenario, such as those increasingly observed during marine heatwaves. While this temperature may exceed average environmental conditions, it remains within the range of short-term maxima recorded in shallow coastal habitats, where polyps naturally occur [29,42].

Indeed, a recent study by Fitt et al. (2025) [42] demonstrated that aposymbiotic *Cassiopea xamachana* polyps from the Florida Keys exhibited no significant changes in tentacle length when exposed to 34 °C. However, exposure to elevated

temperatures of 35–37 °C resulted in a significant reduction in tentacle length, with mortality observed at 36 °C. In contrast, medusae were capable of surviving prolonged exposure to higher thermal conditions (37–39 °C), indicating a greater thermotolerance at later life stages. In the present study, we did not document any polyp mortality, instead at 34 °C polyps exhibited an increase in asexual budding rates compared to control conditions. This elevated budding activity under thermal stress could, in natural environments, contribute to a demographic shift characterized by increased abundance of both polyps and medusae. Such a response may represent an adaptive mechanism in *C. xamachana* populations inhabiting thermally variable environments near the tropical edge of their distribution, such as the Florida Keys.

When studying the effect of this temperature in the symbiotic ephyrae, significant alterations were observed in both morphological characteristics (e.g., bell shape) and symbiont density, suggesting sublethal physiological stress. Notably, the study by Fitt et al. (2025) [42] focused on medusae and did not include earlier developmental stages such as ephyrae, leaving a gap in our understanding of thermal sensitivity during strobilation and early ontogeny.

It is plausible that early life stages, such as ephyrae, exhibit heightened physiological sensitivity to sublethal thermal stress (e.g., at 34 °C), which may not immediately result in mortality but could impair development and reduce fitness in subsequent stages. Cumulative or sustained heat exposure may thus compromise survivorship, developmental trajectories, and reproductive potential. Further investigation is warranted to elucidate the long-term effects of chronic thermal exposure on morphology, physiological function, and viability across early life stages of *C. xamachana*.

## Limitations and future directions

As an initial exploration into how heat stress impacts *C. xamachana*, our study reveals elevated temperatures impact early development. At the same time, we recognize several limitations that offer opportunities for future research. Although our observations highlight disruptions to strobilation, assessing the underlying molecular mechanisms that drive these responses was beyond the scope of this study. Integrating transcriptomic or metabolomic approaches in future work could help identify key regulatory pathways or symbiont-derived cues altered by thermal stress [3,43,44].

While this study briefly examined algal productivity via chlorophyll concentration, more robust data and the measurement of other photophysiological traits will guarantee a better understanding of how symbiont physiology plays a role in host colonization and metamorphosis. Moreover, our design focused on two discrete temperature conditions. Expanding the study to include a broader and more ecologically relevant temperature gradient, as well as varying exposure duration, would help define thresholds and the temporal progression of stress effects with ecological significance. Finally, *C. xamachana* polyps do not undergo strobilation without symbiont colonization. This currently prevents the use of heat-stressed aposymbiotic controls. However, future studies that identify the biochemical triggers of strobilation may enable such experimental manipulations, allowing researchers to disentangle host responses from those of their photosymbionts.

## Conclusions

Our study provides evidence that heat-stressed *Symbiodinium* disrupts *C. xamachana* strobilation, delaying metamorphosis, and, under sustained stress, completely preventing it. Developmental abnormalities, such as inverted bell morphologies and increased symbiont density, suggest altered host-symbiont dynamics, while increased asexual reproduction indicates a stress-induced survival strategy. These findings highlight the broader implications of thermal stress on cnidarian-algal symbiosis. Future research should investigate the biochemical cues driving strobilation, particularly the role of heat-induced metabolite shifts, molecular mechanisms under the physiological changes, and explore whether similar disruptions occur in other photosymbiotic cnidarians. As climate change continues to drive ocean warming, the breakdown of cnidarian-algal symbiosis may have cascading effects on marine ecosystems. By improving our understanding of how *Cassiopea* and its symbionts respond to heat stress, this study contributes to a growing body of research on the resilience and vulnerabilities of photosymbiotic organisms, specifically on the role of host-microbe interactions driving marine invertebrate development in an era of global climate change.

## Supporting information

**S1 Table. Algal cultures chlorophyll t-test data.**
(XLSX)

**S2 Table. Strobilation rate data.**
(XLSX)

**S3 Table. Algal cultures cell count data.**
(XLSX)

**S4 Table. Algal cultures chlorophyll concentration data.**
(XLSX)

**S5 Table. Ephyra algal density data.**
(XLSX)

**S6 Table. T test results comparing in hospite algal cell density.**
(XLSX)

**S7 Table. Asexual budding data.**
(XLSX)

## Acknowledgments

We thank Drs. Pieter Johnson and Nicole Lovenduski from the University of Colorado Boulder for their constructive comments on earlier versions of the manuscript.

## Author contributions

**Conceptualization:** Celeste Robinson, Jingchun Li, Viridiana Avila-Magaña.

**Data curation:** Celeste Robinson, Ruiqi Li.

**Formal analysis:** Celeste Robinson, Ruiqi Li.

**Funding acquisition:** Jingchun Li.

**Investigation:** Celeste Robinson, Jingchun Li, Ruiqi Li, Viridiana Avila-Magaña.

**Methodology:** Celeste Robinson, Viridiana Avila-Magaña.

**Supervision:** Jingchun Li, Ruiqi Li, Viridiana Avila-Magaña.

**Validation:** Ruiqi Li, Viridiana Avila-Magaña.

**Visualization:** Ruiqi Li.

**Writing – original draft:** Celeste Robinson, Ruiqi Li.

**Writing – review & editing:** Jingchun Li, Ruiqi Li, Viridiana Avila-Magaña.

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
