## [Decision Letter · Decision Letter 0]

2 Jul 2025

Dear Dr. Li,

Thank you for submitting your manuscript to PLOS ONE. After careful consideration, we feel that it has merit but does not fully meet PLOS ONE’s publication criteria as it currently stands. Therefore, we invite you to submit a revised version of the manuscript that addresses the points raised during the review process.

We look forward to receiving your revised manuscript.

Kind regards,

Phuping Sucharitakul

Academic Editor

PLOS ONE

Journal Requirements:

[This work is funded by a Packard Fellowship for Science and Engineering (2019–69653) to JL.]. 

[This work is funded by a Packard Fellowship for Science and Engineering (2019–69653) to JL. We thank Drs. Pieter Johnson and Nicole Lovenduski from the University of Colorado Boulder for their constructive comments on earlier versions of the manuscript.]

[This work is funded by a Packard Fellowship for Science and Engineering (2019–69653) to JL.]

[NO authors have competing interests].

Reviewers' comments:

Reviewer's Responses to Questions

**Comments to the Author**

1. Is the manuscript technically sound, and do the data support the conclusions?

Reviewer #1: Partly

2. Has the statistical analysis been performed appropriately and rigorously?

Reviewer #1: N/A

3. Have the authors made all data underlying the findings in their manuscript fully available?

Reviewer #1: Yes

4. Is the manuscript presented in an intelligible fashion and written in standard English?

Reviewer #1: Yes

Reviewer #1: The manuscript PONE-D-25-20894 titled: Heat stress disrupts early development and photosymbiosis in Cassiopea jellyfish has been revised. While the topic is relevant and of broad scientific interest, the presentation of the findings lacks sufficient depth, and a more cautious interpretation of the results is recommended, supported by a clearer and more comprehensive presentation of the data. For these reasons I recommend a major revision.

Major issues

The experimental design is not well presented. You should state how many polyps (pseudo replicates) were in each small well (replicates). You should also better clarify how water changes were performed and how optimal living conditions were maintained throughout the experiment. I recommend including a paragraph in the Discussion section addressing the limitations of this study and outlining how improving the design.

The presentation of the results could be significantly improved. While I do not recall the specific formatting guidelines of Plos One, I strongly recommend separating the Results and Discussion sections to enhance clarity and readability. In the Result section, you should provide numerical values and standard deviation of each obtained result.

The chosen temperature value (34 °C) may be extreme to conduct your experiment. You should consider the possibility that also aposymbionts could show inverted ephyrae due to heat stress only and not due to an impairment of the symbiosis. This possibility should be addressed when interpreting the results and discussing potential confounding factors. It would be beneficial to include a discussion on the natural environmental conditions experienced by the polyps. What potential stressors do they encounter in the wild, and how might these compare to the experimental conditions?

Minor issues were listed below.

Introduction

Line 53. Change in ‘develop into new polyps’

In general, check the word ‘metamorphosis’. You should use it to indicate the formation of a new polyp from the larva/polyp bud. The formation of the ephyra stage is the ‘strobilation’.

Line 68: specify: in Scleractinians instead of in corals.

Line 156. About the sentence: ‘For example, Aurelia aurita strobilates after prolonged cold periods (19), whereas species like Cephea 158 cephea and Rhopilema verrilli respond to elevated temperatures (20,21)’, I suggest considering also papers on strobilation of wild polyps.

Line 158. About the sentence: ‘Unlike these species, Cassiopea relies primarily on colonization by endosymbiotic Symbiodinium to initiate strobilation’, what does it means? Is the strobilation continuous? You should explain in detail this crucial point by summarizing results of the cited papers.

Methods

Line 138: define ‘inverted ephyrae’

Line 154: prevented?

I recommend adding a table to summarize objectives, research questions for each objective, used methods for each research question.

Results

Line 161: please, check grammar of this sentence

Line 196: change in ‘chlorophyll a’

**Do you want your identity to be public for this peer review?** For information about this choice, including consent withdrawal, please see our Privacy Policy

Reviewer #1: No

---

## [Author Response · Author response to Decision Letter 1]

18 Aug 2025

The manuscript PONE-D-25-20894 titled: Heat stress disrupts early development and photosymbiosis in Cassiopea jellyfish has been revised. While the topic is relevant and of broad scientific interest, the presentation of the findings lacks sufficient depth, and a more cautious interpretation of the results is recommended, supported by a clearer and more comprehensive presentation of the data. For these reasons I recommend a major revision.

Response: We thank the reviewers for their constructive feedback. In response, we clarified the experimental design, included numerical values and standard deviations where appropriate, added a new section on study limitations and future directions, expanded the discussion of natural environmental conditions. We believe these changes have strengthened the manuscript.

Major issues

1. The experimental design is not well presented. You should state how many polyps (pseudo replicates) were in each small well (replicates). You should also better clarify how water changes were performed and how optimal living conditions were maintained throughout the experiment.

Response: We appreciate the reviewer’s thoughtful feedback on the clarity of the replicates and maintenance of the 6-well plates. In response, we have clarified the number of polyps per well, rather than just stating the total number of polyps in the entire plate. While there is still some uncertainty, due to there being either 12 or 13 polyps per well, we believe this to be much clearer than simply stating 50 polyps overall. We also added a more detailed account for how the water changes were performed after infection. Finally, we added information regarding the incubators that held constant light and temperature for the culturing and growth of the symbionts and polyps throughout the experiment. Specifically, we clarified the use of two separate incubators to simulate the two different temperatures used in the experiment.

2. I recommend including a paragraph in the Discussion section addressing the limitations of this study and outlining how improving the design.

Response: We appreciate the reviewer’s suggestion. We have added a new “Limitations and Future Directions” section at the end of the Discussion.

Revised text: As an initial exploration into how heat stress impacts C. xamachana, our study reveals elevated temperatures impacts early development. At the same time, we recognize several limitations that offer opportunities for future research. Although our observations highlight disruptions to strobilation, assessing the underlying molecular mechanisms that drive these responses was beyond the scope of this study. Integrating transcriptomic or metabolomic approaches in future work could help identify key regulatory pathways or symbiont-derived cues altered by thermal stress. Moreover, our design focused on two discrete temperature conditions. Expanding the study to include a broader and more ecologically relevant temperature gradient, as well as varying exposure durations, would help define thresholds and the temporal progression of stress effects with greater ecological significance. Finally, C. xamachana polyps do not undergo strobilation without symbiont colonization. This currently prevents the use of heat-stressed aposymbiotic controls. However, future studies that identify the biochemical triggers of strobilation may enable such experimental manipulations, allowing researchers to disentangle host responses from those of the symbionts.

3. The presentation of the results could be significantly improved. While I do not recall the specific formatting guidelines of Plos One, I strongly recommend separating the Results and Discussion sections to enhance clarity and readability.

Response: We appreciate the reviewer’s feedback on improving clarity. While PLOS ONE permits both combined and separate formats, we have opted to retain a combined Results and Discussion section because our findings are straightforward and organized by distinct topics: the effects of heat stress on host strobilation, host development abnormalities, and host reproduction. We believe that presenting and interpreting each topic together improves readability and allows for a more immediate understanding of the implications of each result.

In response to the reviewer’s concern, we have revised the section to ensure that the presentation of results is clearer and more distinct within each thematic subsection. Specifically, we use the first paragraph of each section to summarize all relevant results. We hope this revised structure achieves better clarity while preserving the contextual flow of the discussion.

4. In the Result section, you should provide numerical values and standard deviation of each obtained result.

Response: We have revised the results to add numerical values wherever possible. Numerical values were added in regard to cultured algal density, chlorophyll concentration, ephyra algal cell density, and polyp asexual budding quantifications. Standard deviations were added for polyp asexual budding quantifications and ephyra algal cell density. Standard deviation was not added for chlorophyll concentrations because there were so few measurements taken, so we were able to report them all.

5. The chosen temperature value (34 °C) may be extreme to conduct your experiment. You should consider the possibility that also aposymbionts could show inverted ephyrae due to heat stress only and not due to an impairment of the symbiosis. This possibility should be addressed when interpreting the results and discussing potential confounding factors. It would be beneficial to include a discussion on the natural environmental conditions experienced by the polyps. What potential stressors do they encounter in the wild, and how might these compare to the experimental conditions?

Response: We appreciate the reviewer’s thoughtful comment and agree that 34 °C represents a high thermal condition. This temperature was chosen to simulate an extreme heat stress scenario, such as those increasingly observed during marine heatwaves. While this temperature may exceed average environmental conditions, it remains within the range of short-term maxima recorded in shallow coastal habitats, where polyps naturally occur (Fitt and Costley, 1998, Journal of Experimental Marine Biology and Ecology, Fitt et al., 2025, Oceans).

We fully acknowledge the possibility that the inverted ephyrae observed in the aposymbiotic treatment could result from heat stress alone, rather than solely from the disruption of symbiosis. We have now addressed this potential confounding factor in the revised Results and Discussion section by creating a new subsetction ‘Potential environmental factors impacting Cassiopea life-history’. Furthermore, we have expanded the discussion to include a consideration of natural environmental conditions experienced by polyps in situ. While 34°C is higher than average seasonal temperatures, episodic events such as heatwaves can lead to transient temperature spikes exceeding this threshold. We also suggested that future studies explore long-term exposure to 34°C temperature, and examine combined stressor effects to disentangle the specific impacts of thermal stress versus symbiosis disruption.

Minor issues

Introduction

7. Line 53. Change in ‘develop into new polyps’

Revised.

8. In general, check the word ‘metamorphosis’. You should use it to indicate the formation of a new polyp from the larva/polyp bud. The formation of the ephyra stage is the ‘strobilation’.

We have carefully reviewed the manuscript and corrected the usage of metamorphosis and strobilation.

9. Line 68: specify: in Scleractinians instead of in corals.

Revised.

10. Line 156. About the sentence: ‘For example, Aurelia aurita strobilates after prolonged cold periods (19), whereas species like Cephea 158 cephea and Rhopilema verrilli respond to elevated temperatures (20,21)’, I suggest considering also papers on strobilation of wild polyps.

Response: We agree that understanding strobilation in natural contexts is important. However, due to the challenges of pinpointing environmental triggers in situ, most studies rely on polyps collected from the wild and tested under controlled laboratory conditions. Some studies infer natural triggers through correlations between field observations and environmental parameters, often complemented by lab experiments. In response, we have now cited two relevant studies:

Omori (1995), which monitored seasonal temperature changes in Tokyo Bay and demonstrated that declining temperatures likely trigger strobilation in Aurelia aurita.

Purcell et al. (2009), which analyzed multi-year field data and found that strobilation in Aurelia labiata was associated with higher temperature and high light conditions in natural settings.

These studies illustrate how natural strobilation cues are inferred through correlative approaches. We have revised the manuscript to incorporate these references.

Revised text: Strobilation cues in jellyfish vary widely among species and can be influenced by environmental factors such as temperature, salinity, and light exposure. These patterns are typically inferred by correlating environmental variables with observed strobilation events and are often validated through laboratory studies. For example, Aurelia aurita strobilates after prolonged cold periods (19, Omori 1995), whereas species like Cephea cephea and Rhopilema verrilli respond to elevated temperatures (20, 21). In Aurelia labiata, strobilation has been associated with high-temperature and high-light conditions in natural settings (Purcell et al. 2009).

Added references:

Omori, M., Ishii, H. and Fujinaga, A.I., 1995. Life history strategy of Aurelia aurita (Cnidaria, Scyphomedusae) and its impact on the zooplankton community of Tokyo Bay. ICES Journal of Marine Science, 52(3-4), pp.597-603.

Purcell, J.E., Hoover, R.A. and Schwarck, N.T., 2009. Interannual variation of strobilation by the scyphozoan Aurelia labiata in relation to polyp density, temperature, salinity, and light conditions in situ. Marine Ecology Progress Series, 375, pp.139-149.

11. Line 158. About the sentence: ‘Unlike these species, Cassiopea relies primarily on colonization by endosymbiotic Symbiodinium to initiate strobilation’, what does it means? Is the strobilation continuous? You should explain in detail this crucial point by summarizing results of the cited papers.

We thank the reviewer for helping us to further clarify a key point when presenting the background information regarding the life history of our studied species Cassiopea xamachana. We have added that information in the same section (Line 198).

Revised text: In Cassiopea xamachana, however, this transition is dependent on successful colonization by endosymbiotic Symbiodinium spp., which acts as an essential developmental driver (Hofmann et al., 1978; Colley and Trench, 1985). The presence of these dinoflagellates appears to be a critical environmental and physiological cue required to trigger the transition from the polyp stage (scyphistoma) to the strobila stage, ultimately leading to the release of juvenile medusae (ephyrae). In brief, the transition is induced when symbionts are ingested and phagocytosed by gastrodermal cells, migrating basally within three days (Colley and Trench, 1985; Fitt and Trench, 1983). By day eight, symbiont-containing host cells relocate to the mesoglea and proliferate (Colley and Trench, 1985). Strobilation occurs around three weeks post-colonization, influenced by symbiont species, temperature, and nutrition (Fitt and Costley, 1998).

Methods

12. Line 138: define ‘inverted ephyrae’

We referred as an inverted bell ephyra, based on the morphology observations made in Thayer et al., 2022 “Severe seawater acidification causes a significant reduction in pulse rate, bell diameter, and acute deterioration in feeding apparatus in the scyphozoan medusa Cassiopeia sp.” Invertebr Zool (See figure 5F). We have referenced this paper in the Introduction and changed the term inverted ephyra to inverted bell ephyra throughout the text.

13. Line 154: prevented?

Revised.

14. I recommend adding a table to summarize objectives, research questions for each objective, used methods for each research question.

Response: While we recognize the value of summarizing objectives, research questions, and methods in a table format, we have opted to retain this information in the narrative for clarity and flow. Because the study follows a relatively straightforward rational and method section, we felt that integrating this information into the text maintains cohesion and avoids redundancy. We have revised the Methods sections to make alignment between objectives, questions, and methods more explicit and easier to follow. Specifically, we have reorganized the Methods section so that the order of measurement descriptions now matches the sequence in which the results are presented.

We have also revised the Introduction to clearly state the focus of this study:

Revised text: A combination of morphological observations, symbiont density quantification, and chlorophyll concentration analysis was used to quantitatively and qualitatively assess the effects of heat stress on C. xamachana strobilation timing, ephyrae development and asexual reproduction patterns.

Results

15. Line 161: please, check grammar of this sentence

Revised.

16. Line 196: change in ‘chlorophyll a’

Revised.

---

## [Decision Letter · Decision Letter 1]

14 Sep 2025

Dear Dr. Li,

Thank you for submitting your manuscript to PLOS ONE. After careful consideration, we feel that it has merit but does not fully meet PLOS ONE’s publication criteria as it currently stands. Therefore, we invite you to submit a revised version of the manuscript that addresses the points raised during the review process.

We look forward to receiving your revised manuscript.

Kind regards,

Phuping Sucharitakul

Academic Editor

PLOS ONE

Journal Requirements:

Reviewers' comments:

Reviewer's Responses to Questions

**Comments to the Author**

Reviewer #2: (No Response)

2. Is the manuscript technically sound, and do the data support the conclusions?

Reviewer #2: Partly

3. Has the statistical analysis been performed appropriately and rigorously?

Reviewer #2: N/A

4. Have the authors made all data underlying the findings in their manuscript fully available?

Reviewer #2: Yes

5. Is the manuscript presented in an intelligible fashion and written in standard English?

Reviewer #2: Yes

Reviewer #2: Celeste Robinson and colleagues present the manuscript ‘Heat stress disrupts early development and photosymbiosis in Cassiopea jellyfish’. The authors performed an interesting experiment in which they heat-stressed algal symbionts native to Cassiopea xamachana, and subsequently used the stressed symbionts for the inoculation of aposymbiotic polyps of the jellyfish. In the two heat-stressed group, the authors compare the effects of continuous heat stress (where heat stress is maintained after inoculation of polyps) vs. a recovery phase (where control temperatures are resumed after inoculation), and find interesting stress responses with regard to strobilation in control jellyfish.

While the revision has made some important changes which have strengthened the paper, there are still a few issues that need addressing. I hope my comments below help shape a revision of this interesting work.

Main concerns:

1) The main issue is probably that while the authors show interesting results and patterns, the data are not always robust. For instance, the authors decided not to add standard deviations/errors to chlorophyll a content plots due to having only ‘few measurements’. While the affected results make sense biologically, replication would have been necessary to make this result more robust. This limitation should at least be mentioned in the discussion, and the interpretation should be toned down accordingly – as already mentioned by one of the previous reviewers. I do not feel this point has been sufficiently addressed yet.

2) The discussion is overall well written, but I don’t find it justified that there is not really any attention paid to the potential metabolic role of photosymbiosis in strobilation – it is brushed over very briefly in one sentence, but not really discussed. Considering the importance of symbiotic nutrient cycling in the adult medusa (and many other photosymbiotic Cnidaria), I strongly feel potential host starvation and energy limitation should be discussed in the context of failed strobilation. I understand that the authors are more interested in immunity, but immunity and nutrient cycling are not two isolated entities, but entwined.

3) It is not clear to me whether algal symbionts extracted from ephyra were normalized to anything – ephyra size, weight, protein, etc. - currently, they are only presented as cells per ml. This needs clarification.

4) Fig. 6 is captioned in the text, but missing.

More detailed comments:

Lines 88 and following: Is there any published documentation on the clonal polyp line? I know that the polyp grow quite abundantly under culturing conditions when things are well, but it would be helpful to provide at least a citable resource (reference or protocols.io) if available, for reproducibility. Also, is the San Francisco strain of Artemia commercially available? It would be good to mention the product more specifically.

Lines 109-111: I would like to suggest including Toullec et al. (2024) Microbiome (https://doi.org/10.1186/s40168-023-01738-0) here as well – this is the temperature at which bell pulsation may begin to shift into irregular spasms, and where signs of photosymbiotic breakdown in the medusa are beginning to manifest – irregularities to bell pulsation begins at 34C – in part as a justification.

Lines 121-123: How long did colonization take? In this paragraph, you are only referring to ’12 h post colonization’ – is this really post colonization, or rather post inoculation? Please kindly clarify this section, and be mindful of inoculation vs colonization, which is not the same thing. Please note that in the discussion, proliferation of taken-up cells following infection, and migration (all of which I would consider to be part of the colonization process) is highlighted to start around day 8 post-infection. This needs to be clarified.

Line 123-124: ‘we removed all water from the 6 well plates (…) to remove any culturing media from the symbionts’ – I would remove ‘from the symbionts’, as this can be confusing. I understand now that this is essentially the media in which the polyps were inoculated, but this can be read as if you were mistakenly referring to the polyps as symbionts. Perhaps here you could simply say ‘to remove the spent media after inoculation’?

Lines 137-138: why was host strobilation rate only counted every 3-5 days, why not e.g., daily or at every second day, and why did the time increments differ? How long does strobilation really take? My concern is that strobilation rates might be underestimated at higher temperatures if larvae are overlooked, over 5 days when no quantitation was conducted. Ephyra larvae might die and completely decompose between 3-5 days and go unnoticed.

Lines 143-144: was the homogenized suspension not centrifuged and washed to remove any tissue remains prior to loading the homogenate into the hemocytometer?

Lines 143-144: It is not clear to me whether algal symbionts extracted from ephyra were normalized to anything – ephyra size, weight, protein, etc. - currently, they are only presented as cells per ml – is this supposed to be cells per ephyra? (which is not quantiative). This needs clarification.

Lines 114-115 vs. 144-146: why were methods of algae quantitation switched between inoculation (automated cell counter) and algal counts from ephyra larvae (hemocytometer)? The two different methods have different sensitivities / error rates. Yes, it is not as problematic as counting different sets of algae cultures or different sets of ephyras with different methods, but it seems like an unnecessary switch.

Results and discussion: I just wanted to say that I really appreciate the combined results and discussion format. I know it’s not everybody’s cup of tea, but if the journal supports this format, I am all for it.

Line 168 and following: ‘symbiont cell viability’ – is viability really the best term for density of living algae? How did you discriminate between dead and live algae?

Line 170: please also provide standard errors or deviations for cell density counts – the figure suggests replication, and the point-to-point response suggests this information has been added, but it is not there.

Lines 172-173: are there no replicates for chlorophyll content measurements – both the text and associated fig. 4b give the impression there is just one replicate for each condition and time point? The point-to-point response to one of the previous reviewers suggests ‘few measurements’, but is not clear about not having replication. I would be very careful reporting these data without replication – yes, they do make sense in the biological context, but they are not really robust.

Lines 173-174: ‘symbionts appeared lighter in color compared to dark brown algae cultures’ – this is very confusing – I only understood that two algal cultures are compared here, one in the control, versus one at heat stress when I looked at the photograph in the linked figure. I would urge the authors to be mindful of terminology – ‘symbionts’ suggested to me here that you were referring to freshly isolated symbionts from ephyra, which were contrasted to algal cultures. I would rephrase as follows:

‘Heat-stressed Symbiodinium cultures were also visibly lighter in color compared to the control cultures, supporting lower pigment content.’

Or something along these lines. While not quantitative, this would also lend tentative more support to your chlorophyll measurements, which lack replication.

Lines 236-253: I find the discussion on the roles of photosymbiosis for ephyra development rather underdeveloped, but frankly deserving of taking up more space. Yes, the authors highlighted the effects of aposymbiosis on chemically-induced ephyra, but did not really explain why photosymbiosis is relevant. In line 253, the breakdown of mutualism is mentioned, but really brushed over, before other potential host factors are briefly mentioned. I think the role of metabolism in symbiotic nutrient cycling (which can also be a main driver of symbiont proliferation, via net release of ammonium from the starving host), and its ultimate breakdown alone deserves a few references from both the Cassiopea and coral field, e.g. Toullec et al. 2024 Microbiome, Radecker et al. 2021 PNAS, along with the pioneering work by Ross Cunning. The breakdown of symbiotic nutrient cycling is considered the main driver of cnidarian bleaching and holobiont breakdown today. The algae fix less CO2, and/or retain more organic carbon for their own metabolism, transferring less of it to their host – which essentially starves and has to resort to burning its own proteins once it runs out of carbon. This could potentially be problematic not only for the mature host, but also for more vulnerable early developmental stages, which have very little tissue and probably not much energy storage. This can be summarized and integrated here in a sentence or two and would help make for a more well-rounded discussion.

Line 259: ‘inverted bell-shaped abnormalities’ is rather peculiar phrasing. I suggest rephrasing to ‘abnormalities of the bell, such as inversion’ or ‘abnormalities in shape, such as inverted bells’.

Line 284-286: Fig. 6 (asexual reproduction via budding) is captioned, but missing from the manuscript.

Lines 288-303: this is an interesting discussion, but I find it rather one-sided. I would think that carbon limitation all by itself is a pretty strong argument as well – the algal symbionts provide energy needed for strobilation, if only in the form of ‘junk food’. The ‘checkpoint’ here being: do I even have the energy for strobilation?

Line 341 and following: I would clearly state the absence of replication for chlorophyll a content measurements here as well.

Figures:

Figure 3: ‘strobilated rate’ is unusual; suggestion to replace with ‘strobilation rate’

Figure 4 a: ‘alive algal cell density’ is unusual; suggestion to replace with ‘algal cell densities’ (as in Figure 5b) or ‘densities of viable algal cells’ (IF viability was really assessed) –

Figure 5b: I am not sure I understand the y axis here – why are is there a specific cell density explicitly mentioned in the label of y axis, and what does ‘1, 2, 3’ on the axis then represent? Does this represent 100,000, 200,000, and 300,000 cells? I suggest making the axis label more clear

**Do you want your identity to be public for this peer review?** For information about this choice, including consent withdrawal, please see our Privacy Policy

Reviewer #2: No

---

## [Author Response · Author response to Decision Letter 2]

28 Oct 2025

Please see our attached rebuttal letter.

---

## [Editor Report · Decision Letter 2]

30 Oct 2025

Heat stress disrupts early development and photosymbiosis in Cassiopea jellyfish

PONE-D-25-20894R2

Dear Dr. Li,

We’re pleased to inform you that your manuscript has been judged scientifically suitable for publication and will be formally accepted for publication once it meets all outstanding technical requirements.

Kind regards,

Phuping Sucharitakul

Academic Editor

PLOS ONE
---

## [Editor Report · Acceptance letter]

PONE-D-25-20894R2

PLOS ONE

Dear Dr. Li,

I'm pleased to inform you that your manuscript has been deemed suitable for publication in PLOS ONE. Congratulations! Your manuscript is now being handed over to our production team.

Kind regards,

on behalf of

Dr. Phuping Sucharitakul

Academic Editor

PLOS ONE